# A New Approach to The Synthesis of Polylactide/Polyacrylonitrile Block Copolymers

**DOI:** 10.3390/polym14081529

**Published:** 2022-04-09

**Authors:** Mateusz Grabowski, Bartłomiej Kost, Przemysław Kubisa, Melania Bednarek

**Affiliations:** Centre of Molecular and Macromolecular Studies, Polish Academy of Sciences, Sienkiewicza 112, 90-363 Lodz, Poland; mgrabowski@cbmm.lodz.pl (M.G.); kost@cbmm.lodz.pl (B.K.); pkubisa@cbmm.lodz.pl (P.K.)

**Keywords:** polylactide, copolymer, acrylonitrile, tetraphenylethane, iniferter

## Abstract

As a result of the search for alternatives to the known methods for the synthesis of PLA/vinyl polymer block copolymers, a new approach based on the “iniferter” concept was demonstrated in this article. In this approach, the introduction of a group that was capable of forming radicals and initiating radical polymerization into the polylactide (PLA) chain was conducted. Then, the obtained functional PLA was heated in the presence of a radically polymerizable monomer. The tetraphenylethane (TPE) group was chosen as a group that could dissociate to radicals. PLA with a TPE group in the middle of the chain was prepared in several steps as follows: (1) the synthesis of 4-(2-hydroxyethoxy)benzophenone (HBP-ET); (2) the polymerization of lactide, which was initiated with HBP-ET; and (3) the coupling of HBP-ET chains under UV radiation to form TPE-diET_PLA. A “macroiniferter”, i.e., TPE-diET_PLA, was used to initiate the polymerization of acrylonitrile (AN) by heating substrates at 85 °C. ^1^H and ^13^C NMR and SEC analyses of the products indicated that the triblock copolymer PLA-PAN-PLA formed and thus confirmed the assumed mechanism of the initiation of AN polymerization, which relied on the addition of the radical that formed from TPE (linked with the PLA chain) to the monomer molecule. Copolymerizations were performed with the application of prepared TPE-diET_PLA with three different M_n_’s (1400, 2200, and 3300) and with different AN/PLA ratios, producing copolymers with varied compositions, i.e., with AN/LA ratios in the range of 2.3–11.1 and M_n_’s in the range of 5100–9400. It was shown that the AN/LA ratio in the copolymer was increasable by the applied excess of AN with respect to the PLA macroiniferter in the feed and that more AN monomer was able to be introduced to PLA with shorter chains.

## 1. Introduction

Polylactide (PLA), a renewable polyester that originally had a primary use in biomedical applications and as a packaging material, is now being considered as a substitute for petroleum-derived plastics [1,2]. However, in many applications, PLA has several disadvantages including its relatively low thermal stability and brittleness. One of the methods for modifying the properties of PLA is introducing new chemical units into PLA chains. Copolymers of PLA have been synthesized [3,4,5,6,7,8,9], and they can be applied in various fields, such as tissue engineering [3], drug delivery systems, diagnostic/imaging platforms [4], orthopedic implants, and packaging applications [5], and are widely described in the scientific literature [3,4,5,6,7,8,9].

Among different types of lactide copolymers are block copolymers. Block copolymers with another cyclic monomer are easily synthesized by successive polymerization using the same catalytic system. Problems arise when the other block is built from the polymerization of vinyl monomers according to the radical mechanism. The combination of blocks of PLA and blocks derived from vinyl monomers offers the possibility of modifying the properties of resulting block copolymers within a broad range due to a variety of vinyl monomers and different structures of resulting blocks. There are several approaches to preparing such copolymers, e.g., replacing the end groups of the first block with groups that initiate the polymerization of the monomer forming the second block, using double-acting initiators (initiators containing the ROP initiating group and the group that initiates radical polymerization), or coupling the two previously obtained blocks after introducing groups that react with each other at their ends [9]. Functionalizing PLA to obtain specific functional groups at the ends of macromolecules is often a difficult, multistep process (as polyester, PLA is not very stable under strongly acidic or basic conditions). When polymerizing new monomers, it is necessary to perform purification after the prior synthesis, introduce a new catalyst, and introduce a new solvent. Each of the methods used thus far have certain limitations, so new methods are being sought and developed to replace or simplify the current methods.

In this paper, we describe a method based on the “iniferter” concept [10]. There is a tetraphenylethane (TPE) group among different compounds/groups that acts as an iniferter [10,11]. The TPE group dissociates into free radicals upon heating (see Figure 1). The binding energy between carbon atoms substituted with four phenyl groups is ~200 kJ·mol^−1^ (for (Ph)_2_CH-CH(Ph)_2_); thus, the binding energy is much lower than that of the C-C bond in ethane (377 kJ·mol^−1^) [12].

If a TPE-containing compound (initiator/iniferter) is placed in a medium that contains unsaturated monomers, radicals that are formed upon heating initiate their polymerization [13,14]. According to Braun, when benzopinacol (tetraphenylethanediol, TPED) possessing two tertiary hydroxyl groups is applied as the initiator and the formation of diphenylmethyl radicals has occurred, the initiation proceeds by abstracting the hydrogen radical from the hydroxyl group and adding it to the vinyl monomer [13,14]. However, when -OH groups are blocked, the TPE derivative initiated by an addition to a double bond [15,16].

The TPE group in low-molecular-weight compounds, as well as polymer chains, has been applied as an initiating group in the polymerization of styrene, methyl/butyl/benzyl methacrylate, and acrylonitrile [11,13,14,15,16,17,18,19,20]. Attempts to prepare polyurethanes that contain TPE by reacting tetraphenylethanediol with aromatic diisocyanates have been made and, as claimed by the conductors of such experiments, resulted in “macroiniferters” with several TPE units [11,17,18,19,20]. Some polyurethanes (PUs) have been obtained for which an additional oligodiol, such as poly(tetrahydrofuran), poly(propylene oxide), or poly(ε-caprolactone), was used in PU synthesis [21,22,23]. All polyurethanes served as macroiniferters for the polymerization of vinyl monomers.

We decided to introduce the TPE unit into the PLA chain with the same purpose, i.e., to synthesize the PLA/poly(vinyl monomer) copolymer; however, during our preliminary study, the reaction of isocyanate with tetraphenylethanediol, which contained tertiary -OH groups, was not effective [24]. Other attempts to introduce one TPE unit into the PLA chain by applying TPED as a difunctional initiator of lactide polymerization were also unsuccessful, probably due to the side reactions that occur with benzopinacol in the presence of any catalyst (pinacol rearrangement under acidic conditions [25] or decomposition to benzophenone derivatives under basic conditions) [26]. Thus, we chose another approach, omitting the insufficient reactivity of the sterically hindered tertiary -OH groups in TPED and relying on the preparation of a TPE derivative that contains primary hydroxyl groups that are reactive enough to initiate lactide polymerization.

First, we introduced only one TPE group into the middle of the PLA chain. In our study, the TPE group was introduced into the PLA chain by its formation via the coupling of benzophenone groups that were placed at the polymer chain ends. This type of macroiniferter (PLA-TPE-PLA) should produce triblock copolymer PLA-poly(vinyl monomer)-PLA as a result of heating in the presence of a vinyl monomer. In this study, we examined if PLA copolymers with acrylonitrile (AN) can be obtained by applying a PLA macroiniferter containing TPE units in a specific surrounding. The structures of the obtained copolymers were analyzed, and the mechanism of AN polymerization was suggested. The possibility of regulating the copolymer composition by selecting applied conditions was also examined.

## 2. Materials and Methods

### 2.1. Materials

l,l-Lactide from Purac was recrystallized from 2-propanol, sublimated, and stored under vacuum. 2-Bromoethanol (95%) and 4-hydroxybenzophenone (HBP, 98%), which were both from Sigma-Aldrich (St.Luis, MO, USA/Steinheim, Germany), were used as received, and acrylonitrile (AN, 99%) was passed through an inhibitor removal column filled with silica gel and distilled under vacuum.

Trifluoromethanesulfonic acid from Sigma-Aldrich (St.Luis, MO, USA/Steinheim, Germany), (triflic acid, 99%) was distilled under vacuum. Dichloroethane (DCE, pure p.a.), dimethylformamide (DMF, pure p.a.), and dioxane (pure p.a.) from POCH (Gliwice, Poland) were dried over CaH_2_, distilled before use, and stored over molecular sieves (4 Å). Chloroform (pure p.a.), 2-propanol (pure p.a.), methanol (pure p.a.), ethanol (pure p.a.), diethyl ether (pure p.a.), and hexane (99%), all from POCH (Gliwice, Poland), were used as received. Potassium hydroxide (pure p.a.) was obtained from POCH (Gliwice, Poland), magnesium sulfate (MgSO_4_, 99%) was obtained from Chempur (Piekary Śląskie, Poland) and both were used as received.

### 2.2. Synthesis of the HBP Derivative Containing Primary Hydroxyl Groups (HBP-ET)

The HBP derivative containing primary hydroxyl groups was prepared employing Williamson etherification [27]. A total of 1.68 g (43 mmol) of potassium hydroxide was dissolved in 2.5 mL of distilled water in a beaker. Then, the solution was added to 5 g (25 mmol) of 4-hydroxybenzophenone (HBP) in a round-bottom flask. The resulting yellow reaction mixture was stirred at 70 °C until HBP dissolved. Next, 4 mL (54 mmol) of 2-bromoethanol was added to the syringe, and stirring was continued for 24 h. The aqueous phase was subjected to extraction with chloroform 3 times, and the organic phase was extracted with distilled water 3 times. The combined organic extracts were dried with MgSO_4_. The solvent was evaporated under a vacuum, and the resulting product was recrystallized by being dissolved in 4 mL of ethanol while heated. Then, an excess amount of diethyl ether was added and cooled down, which resulted in growing crystals. The obtained white crystals of HBP-ET, which were dried under vacuum, were obtained with a yield of 50% (>99% pure, ^1^H NMR, GC).

### 2.3. Synthesis of PLA with HBP-ET Units at One Chain End

PLAs were synthesized by the cationic ring-opening polymerization of L-lactide initiated with HBP-ET [28]. The general procedure was as follows (amounts given for HBP-ET_PLA, M_n_ = 1100): LA (3.5 g, 24 mmol) and HBP-ET (0.67 g, 2.7 mmol) were placed in a Schlenk flask, vacuumed, backfilled with nitrogen, and dissolved in 9 mL of DCE, which was added with a syringe via a rubber septum. Next, 0.049 mL (0.56 mmol) of triflic acid was added as a catalyst. The reaction flask was put into an oil bath and stirred at 45 °C for 72 h. After that time, the reaction was stopped by adding CaO, and the reaction mixture was filtered from CaO and precipitated to hexane. The resulting polymer was dried under vacuum; the product was obtained with a yield of ~70%.

### 2.4. Synthesis of PLA with TPE Units in the Middle of the Chain (TPE-diET_PLA)

Polymers were prepared by a radical coupling process that was initiated by UV irradiation [29]. The general procedure was as follows (amounts given for TPE-diET_PLA, M_n_ = 2200): HBP-ET_PLA (3 g) was placed in a quartz flask to which 6 mL of dioxane and 8 mL of 2-propanol were added (both were purged with nitrogen beforehand). The flask was closed with a rubber septum and purged with nitrogen (with a needle) for 15 min. Then, the flask content was slightly heated until it dissolved. The homogenous solution was exposed to ultraviolet light (365 nm) for 24 h. The resulting solution was evaporated under reduced pressure and dried. Subsequently, the polymer was dissolved in DCE and precipitated to hexane, decanted, and dried under vacuum. The resulting polymer was obtained with a yield of ~90%.

### 2.5. Polymerization of AN Using TPE-diET_PLA as a Macroiniferter

The synthesis of the TPE-diET_PLA/PAN copolymer was performed using the iniferter concept [10]. TPE-diET_PLA (0.3 g, 0.13 mmol) was placed in a Schlenk flask, vacuumed, and filled with nitrogen. Next, 0.32 mL of AN (4.7 mmol) was added to the reaction flask through a rubber septum. The solution was degassed by freeze–pump–thaw cycles. The flask was filled with nitrogen and placed and heated in an oil bath at 85 °C for 24 h with stirring. The resulting polymer was precipitated to cold methanol, centrifuged, and dried under reduced pressure.

### 2.6. Instrumental Methods

^1^H NMR spectra of the synthetized PLA polymers were recorded in CDCl_3_ and in DMSO-d6 using a Bruker Avance 400 Neo instrument (Bruker, Billerica, MA, USA) operating at 400 MHz.

Size exclusion chromatography (SEC) was performed using an Agilent Pump 1100 Series with an Agilent G1322A Degasser (Santa Carla, CA, USA) for analyses in DCM and a Shimadzu Pump LC-20AD with Shimadzu DGU-20A5 Degasser (Kioto, Prefektura Kioto, Japan) for analyses in DMF. For both systems, a set of two PLgel 5 μm mixed-C columns and a Wyatt Optilab REX interferometric refractometer (Dernbach, Germany) were used. A flow rate of 0.8 mL·min^−1^ was applied for the eluent, and analyses were performed at room temperature in DCM and at 40 °C in DMF. The systems were calibrated with polystyrene standards.

Fourier transform infrared spectroscopy (FTIR) measurements were performed on a Thermo Scientific Nicolet 6700 instrument with an attenuated total reflectance (ATR) GoldenGate accessory (Waltham, MA, USA) and with a deuterated triglycine sulfate (DTGS) detector. The spectra were obtained by adding 64 scans at a 2 cm^−1^ resolution.

UV coupling was performed in a photochemical reactor (RayonetRPR-200, Southern New England, Brandford, CT, USA) equipped with 12 UV lamps (λ = 365 nm) and a magnetic stirrer.

## 3. Results and Discussion

### 3.1. Synthesis of a PLA Containing TPE Group in the Middle of the Chain

The synthesis of a PLA-based macroiniferter containing an active group in the middle of the PLA chain was performed using the following steps: (1) functionalization of 4-hydroxybenzophenone with 2-bromoethanol, which led to the introduction of a primary -OH group into the benzophenone derivative 4-(2-hydroxyethoxy)benzophenone (HBP-ET), (2) initiation of lactide polymerization with HBP-ET (HBP-ET_PLA), and (3) coupling of HBP-ET_PLA to the TPE-diET_PLA polymer. The corresponding reaction scheme is presented in Figure 2.

The successful functionalization of 4-hydroxybenzophenone to 4-(2-hydroxyethoxy)benzophenone (HBP-ET) was confirmed by ^1^H NMR analysis (see Appendix A). HBP-ET containing primary hydroxyl groups was used for the initiation of l,l-lactide polymerization, which was catalyzed by triflic acid. PLAs with three different molecular weights were obtained. The molecular weights were M_n_ = ~700, 1100, and 1600 (M_n_ determined by ^1^H NMR, Figure 3, Appendix A). These polymers possessed a benzophenone moiety at one chain end and were dissolved in a solvent that contained isopropanol as a hydrogen atom donor, and they were subjected to UV radiation, which resulted in the coupling of two PLA chains and the formation of a tetraphenylethane group in the middle of the polymer chain. The expected structure of the coupled PLAs was confirmed by ^1^H NMR, ^13^C NMR, ^1^H^13^C-HMBC correlation, and FT IR analyses (Figure 3, Appendix A).

### 3.2. Polymerization of Acrylonitrile Initiated by TPE-diET_PLA

Different vinyl monomers can be polymerized by the radicals that form from the dissociation of the TPE group; however, for our study, we chose a monomer that was not thermally polymerizable. Acrylonitrile was determined to be stable at 85 °C, which is a sufficient temperature for achieving the reasonable decomposition of TPE.

To obtain PLA/PAN copolymers, TPE-diET_PLA was placed in a Schlenk tube, dissolved in a solvent (DMF or DMSO for the kinetic experiment), and mixed with acrylonitrile; after deoxygenation, the flask was immersed in a heated oil bath at 85 °C (a higher temperature can result in faster polymerization but also some side products). After a predetermined time, the solvents were evaporated, and the product was analyzed by ^1^H NMR (some polymers were also analyzed by ^13^C NMR) and SEC analyses. SEC curves revealed the additional presence of unreacted PLA homopolymers in a sample; thus, the polymerization product was precipitated with methanol and analyzed again (Polylactide with a small molecular weight is soluble in methanol). Figure 4 presents ^1^H NMR spectra and SEC curves of the precipitated product from the AN polymerization initiated by TPE -diET_PLA with a relatively low M_n_.

In the ^1^H NMR spectrum, all the signals that were expected for the TPE-diET_PLA/PAN copolymer were present. In comparison with the spectrum for TPE-diET_PLA, new signals corresponding to methylene at 2.0–2.25 ppm and methine protons at 3.0–3.25 ppm from the formed PAN block appeared. On the basis of corresponding integrals, the AN/LA and TPE/LA ratios were calculated (the method of calculation is explained in the SI). For the copolymer presented in Figure 4, these ratios were equal to 11.1 and 0.153, respectively, and in the starting reaction mixture, they were equal to 4.2 and 0.157, respectively. The change in the AN/LA ratio was due to the noticeable fractionation (approximately 60% was removed) of the solid product during its precipitation and the removal of unreacted PLA. The SEC curve of the polymerization product indicated an increase in molecular weight in comparison with the starting TPE-diET_PLA.

Signals corresponding to TPE units in the PLA chain in the range of 6.65–7.4 ppm were visible, although their shape (multiplicity) differed from those present in the spectrum of TPE-diET_PLA. New signals appeared at higher chemical shifts that resembled those characteristic of 4-hydroxybenzophenone phenyl protons, which suggested the presence of diphenyl methyl groups that were formed by the decomposition of TPE groups. Successive decomposition of the TPE groups with increasing conversion of the AN monomer was demonstrated by kinetic experiments. Figure 5 presents the ^1^H NMR spectra of the reaction mixture recorded at different times of TPE-diET_PLA (M_n_ = 2200 1:1 wt. ratio of AN to TPE_PLA) heating in DMSO as a solvent.

Figure 5 shows that the intensities of signals that corresponded to protons in the AN monomer in the range of 5.8–6.4 ppm decreased and the intensities of signals corresponding to formed PAN units increased with the heating time. As mentioned above, with increasing heating time, the successive decomposition of TPE groups was observed. Appropriate calculations (which are presented in the SI) allowed us to determine the AN conversion and percent of TPE group decomposition. Figure 6 presents kinetic plots for the AN polymerization in comparison with the decomposition of the TPE group.

Irreversible TPE decomposition proceeded successively during heating along with initiation of the polymerization and consumption of the AN monomer. The conversion of AN almost stabilized after 15 h at a level of ~85% (in DMSO-d_6_), which was when the irreversible decomposition of the TPE groups was ~60%. By prolonging the reaction time, further TPE decomposition could have been achieved; however, the reaction was stopped to avoid side reactions (which proceed with a high AN conversion, leading to a deteriorated quality of the product, as is shown later).

^1^H NMR analysis of the reaction mixture and obtained product confirmed that the polymerization of AN was initiated by the TPE group present in the PLA chain. SEC analysis indicated the formation of a (co)polymer with a molecular weight higher than that of the initiating TPE-diET_PLA (shifting of the SEC curve, Figure 4) and similar to that calculated from the ^1^H NMR spectrum with the assumption of a triblock copolymer structure (values shown in Table 1). However, some doubts related to the mechanism of vinyl monomer polymerization that was previously postulated by Braun (see Introduction, [13,14]) should be dispelled. Braun claimed that if TPED possessing tertiary hydroxyl groups is used as an initiator of unsaturated monomer polymerization, the growing polymer chain will not contain the TPED fragment as an end group. In our work, although the TPE group was linked to the PLA chain, it still contained tertiary -OH groups, and the same mechanism that was postulated by Braun could have occurred and led to a PAN homopolymer instead of the copolymer with PLA. These two mechanisms are presented in Figure 7.

An analysis of the product microstructure based on a ^13^C NMR spectrum provided further information about the polymerization mechanism. Additional polymerization of AN initiated with TPE_PLA 1400, with the AN/LA wt. ratio in the feed equal to 0.5, was performed, and the product was separated after 5 h by the evaporation of unreacted AN and solvents (this procedure allowed us to prevent the formed products from changing in the analyzed sample). Figure 8 presents ^13^C NMR spectra for the polymerization product, together with the spectra for the initial PLAs for comparison. In the spectrum for the formed product (expected PLA/AN copolymer), a signal that was assigned to a quaternary carbon atom from the decomposed TPE unit attached to the AN unit at ~82 ppm was visible. On the other hand, the signal corresponding to the carbonyl carbon characteristic of the benzophenone moiety at ~195 ppm was not present. The mechanism of some vinyl monomers’ polymerization initiated by TPED, suggested by Braun, should lead to the formation of AN homopolymer (without a TPE fragment) with simultaneous benzophenone formation. As a carbonyl group that corresponded to benzophenone was not formed in the reaction we studied, it was concluded that the polymerization did not proceed according to Braun’s scheme but rather by the simple addition of the formed (TPE-diET_PLA)/2 radical to the AN monomer. The ^13^C NMR analysis delivered unequivocal evidence that a block copolymer of PLA and PAN was formed, in which both blocks were linked through half of the TPE-diET unit. The most probable situation was that growing radicals recombined (or a growing radical combined with the (TPE-diET_PLA)/2 radical); thus, it was assumed that the triblock copolymer PLA-PAN-PLA was obtained.

To show that it is possible to obtain PLA-PAN-PLA copolymers with varying lengths of PLA and PAN blocks by manipulating the molecular weight of the initial PLA and the AN to PLA ratio in the feed, several polymerizations were performed with three different TPE-diET_PLAs. ^1^H NMR spectra of the products, as well as SEC traces, are shown in the Supporting Information (Appendix A), and the product composition and molecular weights are presented in Table 1.

The conversion of AN in the range of 62–73% was achieved within a 24 h reaction time. A prolongation in the reaction time results in a higher conversion, although the polymerization rate considerably decreases (this is shown by the kinetic experiment shown earlier in this contribution). The applied excess of AN with respect to the PLA polymer (No. 2–4 in Table 1) influences the length of the AN block. On the other hand, the ratio of AN to LA units may also be regulated by the molecular weight of the PLA containing TPE unit. In the case of reactions with the same excess of AN, a higher conversion was achieved when the TPE-diET_PLA with a lower M_n_ was used as a “macroiniferter”, which may be explained by the better accessibility of the TPE group. However, it should be remembered that this ratio depends on the conversion of the AN monomer and product separation because its precipitation leads to the removal of unreacted PLA (only PLA was found in the solution after precipitation).

The TPE/LA ratio varied to a lesser extent and was close to the ratio in the initial TPE-diET_PLA (differences were due to polymer fractionation). The fact that the molecular weights of the obtained copolymers determined by SEC (with the awareness of the error resulting from the assumed method of counting) were close to those calculated from ^1^H NMR spectra with the assumption of triblock copolymer formation supports the thesis that triblock copolymers, i.e., PLA-PAN-PLA, were formed by the recombination of two growing radicals, i.e., PLA-PAN•.

## 4. Conclusions

Failed initial attempts to introduce the TPE iniferter group into the PLA chain through the use of commercial TPED led us to develop a new synthetic methodology. This approach is based on the modification of 4-hydroxybenzophenone, the polymerization of lactide initiated by primary hydroxyl groups, and finally, the coupling of benzophenone groups that are placed at polymer chain ends. The expected structure of the resulting macroiniferter was confirmed by detailed analysis using NMR and FTIR spectroscopy and SEC. The application of the TPE-diET_PLA low-molecular-weight macroiniferter for the initiation of acrylonitrile polymerization resulted in a PLA-PAN-PLA triblock copolymer, which was demonstrated by ^1^H and ^13^C NMR as well as SEC analyses of the synthesized products. The acquisition of block copolymers confirmed the expected AN initiation mechanism, namely, through TPE dissociation (in the middle of the PLA chain) and then the addition of the resulting radicals to vinyl monomer molecules.

It was shown that the composition of copolymers was regulated by the manipulation of the macroiniferter length, substrate ratio, and polymerization conditions.

The presented approach may be applied for monomers other than those described in the present contribution; thus, it is a suitable method for the preparation of block copolymers from monomers polymerized by different mechanisms.

## Figures and Tables

**Figure 1 polymers-14-01529-f001:**
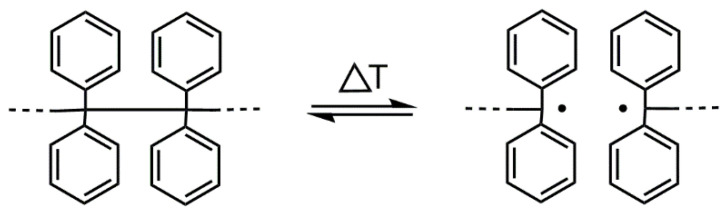
Reversible thermal dissociation of the TPE group.

**Figure 2 polymers-14-01529-f002:**
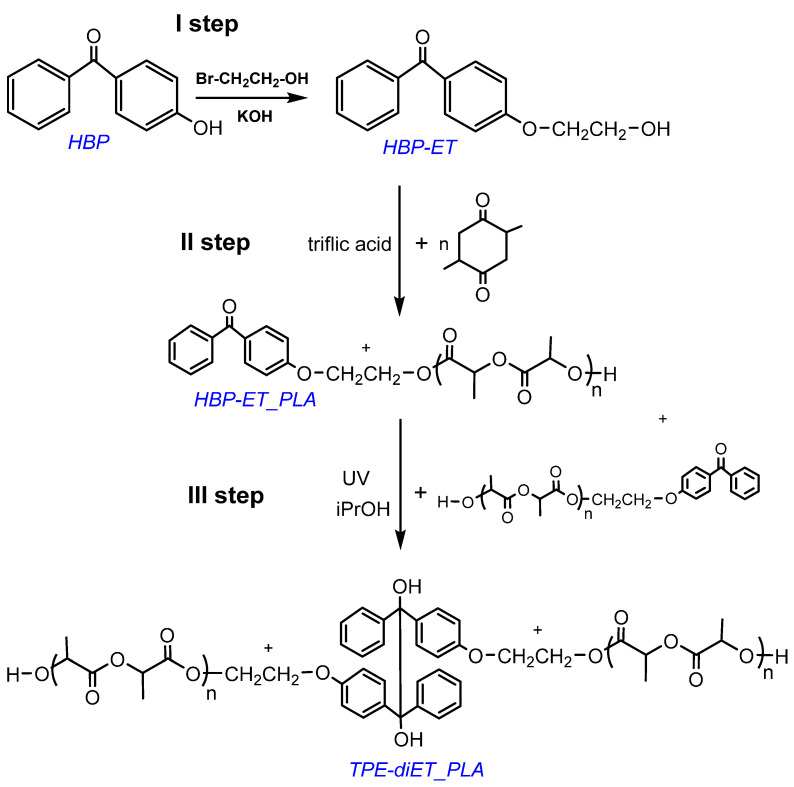
Scheme for the synthesis of PLA with one TPE group in the middle of the chain.

**Figure 3 polymers-14-01529-f003:**
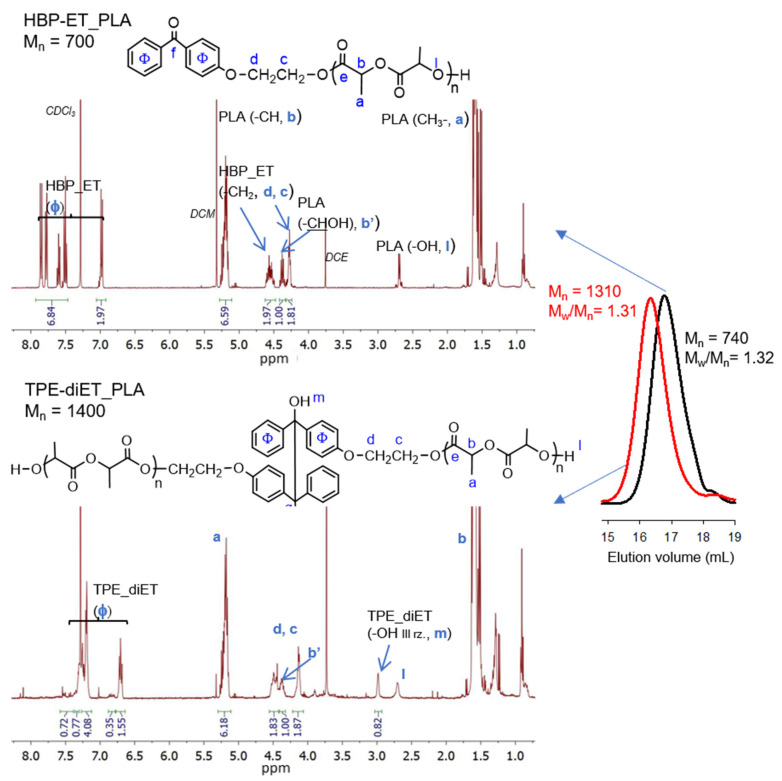
**^1^**H NMR spectra (CDCl_3_) and SEC curves (DCM) of HBP-ET_PLA, M_n_ =700 (top, SEC black line) and TPE-diET_PLA, M_n_ = 1400 (bottom, SEC red line).

**Figure 4 polymers-14-01529-f004:**
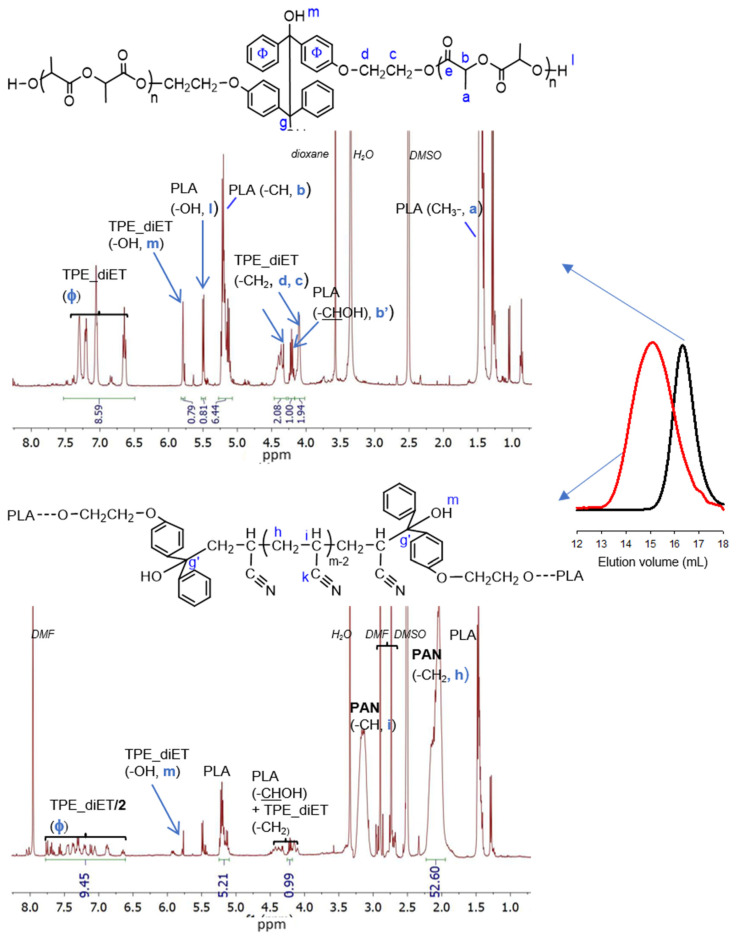
^1^H NMR spectra (DMSO-d6) and SEC curves (DMF) of TPE-diET_PLA, M_n_ = 1400 (on the top, SEC black line), and the precipitated product of the AN polymerization initiated with TPE-diET_PLA; m_AN_/m_PLA_ in feed =1; 24 h (on the bottom, SEC red line).

**Figure 5 polymers-14-01529-f005:**
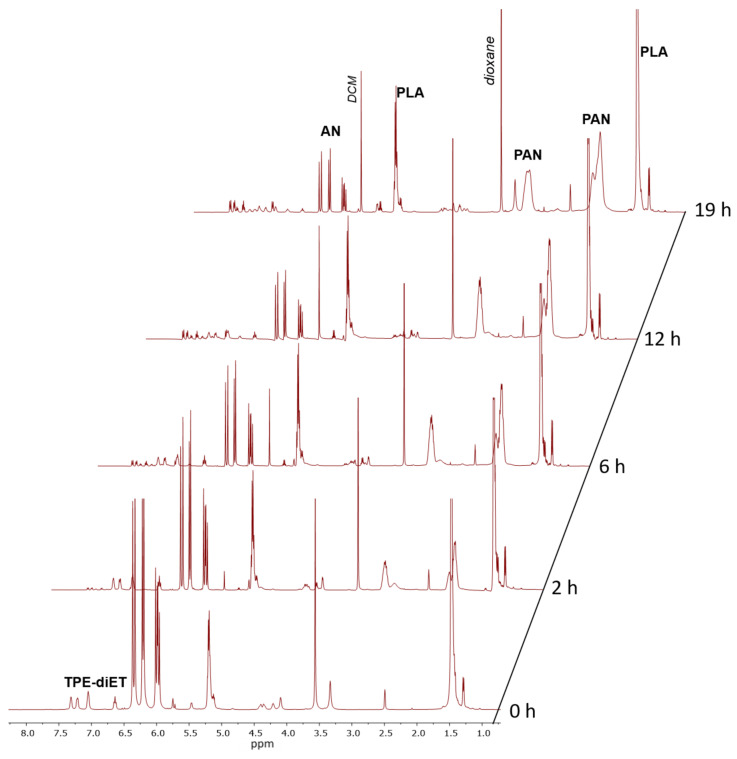
^1^H NMR spectra of the solution of TPE-diET_PLA reacting with acrylonitrile in DMSO-d_6_ at 85 °C recorded at different times.

**Figure 6 polymers-14-01529-f006:**
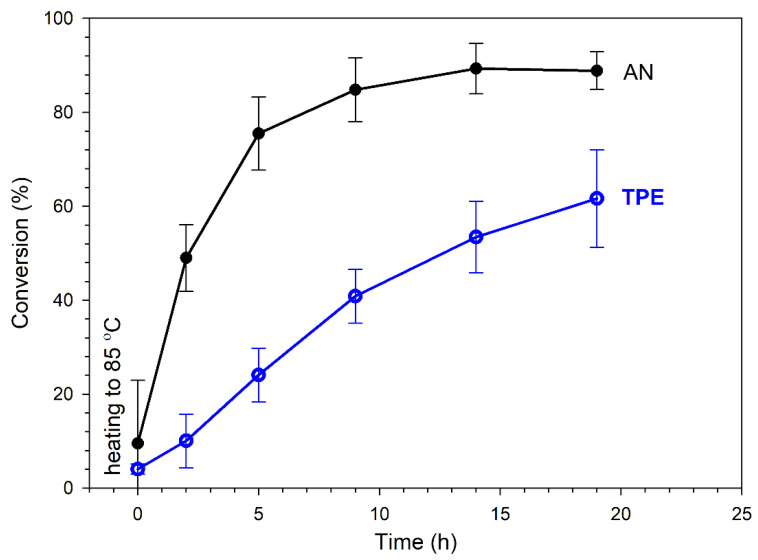
Conversion of AN and TPE groups decomposition as a function of heating time (reaction in an NMR tube in DMSO-d_6_); plots based on average values of three measurements.

**Figure 7 polymers-14-01529-f007:**
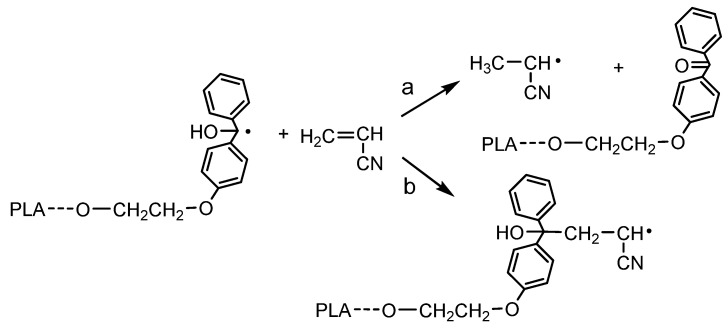
Two possible mechanisms for the initiation of AN polymerization by radicals formed from TPE-diET_PLA: (**a**) according to Braun vs. (**b**) proceeding by the addition of the (hyroxy)diphenylmethyl radical connected with the PLA chain to the AN monomer.

**Figure 8 polymers-14-01529-f008:**
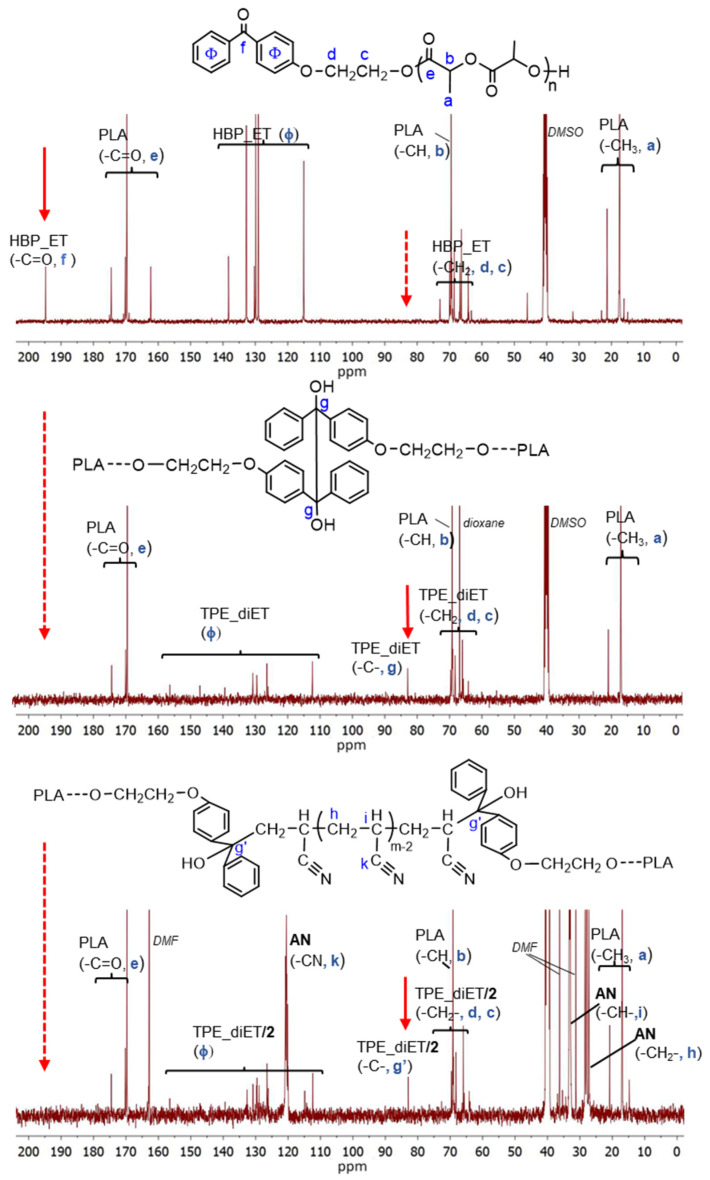
^13^C NMR spectra of HBP-ET_PLA (before coupling), TPE-diET_PLA (after coupling), and the product of the AN polymerization initiated with TPE-diET_PLA (TPE/PAN copolymer).

**Table 1 polymers-14-01529-t001:** Polymerization of acrylonitrile at 85 °C for 24 h.

TPE_PLA		Polymerization No.	AN/LA mol Ratio in Feed (wt. ratio)	Conversion of AN, % ^2^	PLA-PAN-PLA Copolymers ^3^		
M_n_ ^1^ g/mol	TPE/LA				AN/LA ^1^	TPE/LA ^1^	M_n_^1^ (SEC) ^4^ g/mol
1400	0.157	1	4.2 (1)	73	11.1	0.153	5130 (4250)
2200	0.084	2	1.7 (0.5)	65	2.5	0.049	5760 (4580)
		3	3.2 (1)	69	3.45	0.060	5670 (4780)
		4	7.0 (2)	71	9.9	0.071	9370 (8500)
3300	0.054	5	3.5 (1)	62 81_4 days 89_7 days	2.3 N.d. ^5^ ”	0.050 N.d. ^5^ ”	6070 (5850) Bimodal SEC “

^1^ The M_n_ of TPE_PLA (=TPE-diET_PLA), as well as the M_n_, AN/LA, and TP/LA ratios in copolymers, were calculated on the basis of the intensity of appropriate signals in the ^1^H NMR spectra of the precipitated products (the method of calculation is presented in SI). ^2^ Conversion was determined on the basis of the ^1^H NMR spectrum of the reaction mixture. ^3^ PLA-PAN-PLA copolymers were analyzed after being precipitated to methanol, which resulted in the fractionation of the starting product; the precipitated polymer accounted for approximately 40% (No. 1), 65% (No. 3), and 50% (No. 5) of the starting product. ^4^ The approximate M_n_ from SEC analysis in DMF, shown for comparison, was calculated as M_n_ = M_n_ from SEC measurement with PS standards divided by 6.5 (for sample 1) or 6 (for samples 2–5). Factors “6.5” or “6” were found by comparing M_n_’s determined for HBP-diET_PLA in DCM and in DMF. ^5^ Not determined.

## Data Availability

The data presented in this study are available in the article text and Appendix A.

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
