# Peer review of "A New Approach to The Synthesis of Polylactide/Polyacrylonitrile Block Copolymers"

_polymers, 2022, doi:10.3390/polym14081529_

Round 1
Reviewer 1 Report
polymers-1642761
The article: “A new approach to the synthesis of polylactide/polyacrylonitrile block copolymers” by M. Grabowski et al. describes the synthesis of copolymers of PLA and PAN by the combination of ring-opening polymerization and radical polymerization. The authors used a macroinitiator or “macroiniferter” consisting of PLA-TPE to initiate the polymerization of AN. The synthesized polymers were in a low range of molecular weights and were characterized by SEC, 1H NMR, 13C NMR, and FT-IR.
Although the topic of the research seems interesting, the article's presentation is not in a satisfactory way. The MS needs serious editing in English and the preparation and discussion of the results. Moreover, the supporting information was not available to the reviewer, making it difficult to follow the results and discussion section. Unfortunately, I suggest that it is not suitable for publication in Polymers in its current form. The following comments were raised while reading the MS:
- What is the yield of TPE-diET-PLA polymer?
- In Figure 2, no need to include all this information in the caption. These details should be included in the results and discussion section, where there is no discussion regarding the synthesis of HPB-ET-PLA polymers.
- 3: The authors need to clarify in the caption the SEC traces that are presented in the figure. Moreover, complete assignment of the protons should be provided in both spectra. Which protons of PLA are referring to 1.5-1.6 ppm or at 5.3 ppm?
- 4: Again, the authors should assign the protons of each peak. Which spectrum corresponds to TPE-PLA and which to the copolymer with PAN?
- In Table 1, the authors should mention the units below each column.
- Do the authors have proof regarding the possible mechanisms of the initiation of AN presented in Fig. 7? Since the study of these copolymers is based on low MWs, is it possible to study them through MALDI-TOF?
- Have the authors tried to synthesize higher MW copolymers?
Author Response
Please see the attachement.

Reviewer 2 Report
- The abstract is written the same with an introduction and is too general. Must include some important data and explain the methodology, characterization, and some important data.
- The English is too weak and must be improved by a native English.
- “Copolymers of polylactide are being synthesized, they find the application in various fields and are widely described in the scientific literature [3–9]“.My suggestion is instead of this general sentence and 7 references, make the sentence separately clear with the applications and its relevant references.
- Figure 1 is not showing any important information. Write down the name of each compound on or around it. Also, delta is for how much heating temperature?
- The last paragraph of the introduction belongs to the aim of the study, why reference is coming in between? Reference 27!!!
- All the methodologies need references.
- There is no statistical analysis in the manuscript!!
- There are 27 references and 13 of them are too old. Why are the authors referring to such old references? Please try to update the references with newly published papers.
- As a suggestion, the authors can use the following references in the manuscript:
Sabbagh, F., Muhamad, I. I., Nazari, Z., Mobini, P., & Khatir, N. M. (2018). Investigation of acyclovir-loaded, acrylamide-based hydrogels for potential use as vaginal ring. Materials Today Communications, 16, 274-280.
Reviewer 3 Report
In this paper, the authors reported a novel synthesis procedure of a block copolymer, PLA-TPE-PLA, to develop the synthesis method of functionalized polyester. The manuscript contains original contributions, developing the polymerization and functionalization process of block copolymer and characterizing the product block copolymers, for the research fields of macromolecule synthesis. However, the manuscript has insufficient explanations of figures and discussion of the results. The authors should consider the following comments and revise the manuscript before publishing the manuscript on “polymers”.
- In Results and discussion part, pages 4,5, Fig. 2 and 3, please explain Figure 3 in the main text, not only figure caption of Figure 2, in detail. In addition, if the author shows the data of SEC in Figure 3, please add the description in the figure caption.
- In Results and discussion part, page 7, Fig. 4, please add the description or legend of each graph in the figure and figure caption. It is difficult to understand the data for readers.
- In Results and discussion part, page 8, Fig. 5, is there any effect of the reaction temperature on the synthesis of TPE-diET_PLA?
- In Results and discussion part, page 9, Fig. 6, which peak of NMR was used for calculation of conversion rate? Why is the reaction trend of AN and TPE different? Does the author estimate the rate of both reactions?
- In Results and discussion part, page 12, Table 1, please explain the relationship between the mixing ratio of AN/LA as precursor materials and the ratio of AN/LE of products. Why is the ratio change unproportional?
- In Conclusions part, page 12, the content of the first paragraph can be mentioned in Introduction part or discussed in Results and discussion part. The author can summarize the Conclusions part.
- There are several mistakes in the text and table mentioned following.
  7-1 In Introduction part, page 1, please revise the mistake, “~ monomer offers. a possibility ~” should be “~ monomer offers a possibility ~”.
  7-2 In Table 1, please revise the mistake, “0,049” should be “0,049” at the column of TPE/LA on PLA-PAN-PLA copolymers.
Round 2
Reviewer 1 Report
The authors revised the manuscript in a satisfactory way and based on the reviewer's comments. I suggest that it can be published in Polymers in its current form.
Reviewer 3 Report
The author carefully and well revised the manuscript. The manuscript can be accepted without further revision.